# How Fine-Tuning Allows for Effective Meta-Learning

**Kurtland Chua**
Princeton University
kchua@princeton.edu

**Qi Lei**
Princeton University
qilei@princeton.edu

**Jason D. Lee**
Princeton University
jasonlee@princeton.edu

## Abstract

Representation learning has served as a key tool for meta-learning, enabling rapid learning of new tasks. Recent works like MAML learn task-specific representations by finding an initial representation requiring minimal per-task adaptation (*i.e.* a fine-tuning-based objective). We present a theoretical framework for analyzing a MAML-like algorithm, assuming all available tasks require *approximately* the same representation. We then provide risk bounds on predictors found by fine-tuning via gradient descent, demonstrating that the method provably leverages the shared structure. We illustrate these bounds in the logistic regression and neural network settings. In contrast, we establish settings where learning one representation for all tasks (*i.e.* using a "frozen representation" objective) fails. Notably, *any such algorithm* cannot outperform directly learning the target task with no other information, in the worst case. This separation underscores the benefit of fine-tuning-based over "frozen representation" objectives in few-shot learning.

## 1 Introduction

Meta-learning (Thrun & Pratt, 2012) has emerged as an essential tool for adapting prior knowledge to new tasks under data and computational constraints. In this context, a meta-learner has access to related source tasks from a shared environment. The learner aims to uncover inductive biases from the source tasks to reduce the sample/computational complexity of learning new tasks from the same environment. A common approach is representation learning (Bengio et al., 2013), *i.e.* learning a feature extractor from the source tasks. At test time, a learner adapts to a new task by fine-tuning the representation and retraining the final layer(s) (see, *e.g.*, prototype networks (Snell et al., 2017)). Substantial improvements over directly learning a single task has been shown in few-shot learning (Antoniou et al., 2018), a natural setting in many applications including reinforcement learning (Mendonca et al., 2019; Finn et al., 2017), computer vision (Nichol et al., 2018), federated learning (McMahan et al., 2017) and robotics (Al-Shedivat et al., 2017).

The empirical success of representation learning has led to an increased interest in theoretical analyses of underlying phenomena. Recent work assumes an explicitly shared representation across tasks (Du et al., 2020; Tripuraneni et al., 2020b,a; Saunshi et al., 2020; Balcan et al., 2015). For instance, Du et al. (2020) prove a generalization bound consisting of a representation error term and an estimation error term. However, without fine-tuning the whole network, the representation error accumulates to the target task and is *irreducible*, even with an infinitely large target task dataset. This result is consistent with empirical findings, which suggest that fine-tuning the whole network provides substantial performance gains, compared to just learning the final linear layer (Chen et al., 2020; Salman et al., 2020). Due to the lack of (representation) fine-tuning while training on the source tasks, we refer to methods above as making use of "frozen representation" objectives.

Linear separability of tasks on the same features poses additional problems, being an unrealistic assumption when transferring knowledge to other domains (*e.g.* from ImageNet to medical images (Raghu et al., 2019)). Therefore, we consider a more realistic setting where the available tasks only *approximately* share the same representation. In this setting, we propose a theoretical framework for

35th Conference on Neural Information Processing Systems (NeurIPS 2021).

analyzing the sample complexity of fine-tuning, using an initial representation from a MAML-like algorithm. We show that fine-tuning quickly adapts to new tasks, requiring fewer samples in certain cases compared to methods using "frozen representation" objectives. To the best of our knowledge, no prior theoretical studies exist beyond fine-tuning a linear model (Denevi et al., 2018; Konobeev et al., 2020; Collins et al., 2020a; Lee et al., 2020) or only the task-specific layers (Du et al., 2020; Tripuraneni et al., 2020b,a; Mu et al., 2020). Tripuraneni et al. (2020b), in particular, acknowledge that their work does not incorporate representation fine-tuning, leaving such analysis to future work. It is this gap which we now seek to address.

The following outlines this paper and its contributions:

In **Section 2**, we outline the general setting and our overall assumptions. Additionally, we introduce ADAPTREP, the representative fine-tuning-based algorithm we analyze in this work. As a baseline, we also formally define FROZENREP, which makes use of a "frozen representation" objective.

In **Section 3**, we provide an in-depth analysis of the ($d$-dimensional) linear representation setting, when learning a $k$-dimensional representation. First, we show that ADAPTREP achieves rates of

$$r_{\text{source}} = O\left(\frac{kd}{n_{\text{S}}T} + \frac{k}{n_{\text{S}}} + \delta_0\sqrt{\frac{\text{tr}\,\Sigma}{n_{\text{S}}}}\right) \quad \text{and} \quad r_{\text{target}} = O\left(\frac{k}{n_{\text{T}}} + \delta_0\sqrt{\frac{\text{tr}\,\Sigma}{n_{\text{T}}}} + r_{source}\right)$$

on the source and target tasks. Here, $n_{\text{S}}$ and $n_{\text{T}}$ are the number of available samples, $T$ is the number of source tasks, $\delta_0$ is the norm of the representation change, and $\Sigma$ is the input covariance. Thus, fine-tuning can handle the misspecified setting. In contrast, FROZENREP has a minimax rate of $\Omega(d/n_{\text{T}})$ on the target task under certain task distributions, matching standard linear regression. We provide a formal construction and an experimental verification of the gap in **Section C**.

In **Section 4**, we extend the analysis to general function classes. We provide risk bounds of the form

$$\varepsilon_{\text{OPT}} + \varepsilon_{\text{EST}} + \varepsilon_{\text{REPR}},$$

where $\varepsilon_{\text{OPT}}, \varepsilon_{\text{EST}}, \varepsilon_{\text{REPR}}$ are the optimization, estimation, and representation errors, respectively.

The *optimization error* $\varepsilon_{\text{OPT}}$ quantifies the error from using approximate minima found during optimization. To control $\varepsilon_{\text{OPT}}$, our analysis accounts for nonconvexity introduced by representation fine-tuning, and is presented as a self-contained result in **Section H**.

The *estimation error* $\varepsilon_{\text{EST}}$ stems from fine-tuning to the target task with finite $n_{\text{T}}$ samples. It therefore scales with $1/\sqrt{n_{\text{T}}}$ and is controlled by the complexity of the target fine-tuning set.

The *representation error* $\varepsilon_{\text{REPR}}$ is the error incurred by representation learning on the source tasks while adapting to them. It consists of two terms: one scaling as $1/\sqrt{n_{\text{S}}T}$ for learning an initialization using all $T$ source tasks, and the other scaling as $1/\sqrt{n_{\text{S}}}$ for learning task-specific adaptations.

In **Section 5**, we instantiate our guarantees in the two-layer neural network setting. Additionally, we provide an analysis for logistic regression in **Section F**. Furthermore, we extend the linear hard case to a nonlinear setting in **Section G**.

## 1.1 Related Work

The empirical success of MAML (Finn et al., 2017), and meta-learning in general, has elicited further theoretical analysis from both statistical and optimization perspectives. A flurry of work engages in developing more efficient and theoretically-sound optimization algorithms (Antoniou et al., 2018; Nichol et al., 2018; Li et al., 2017) or convergence analyses (Fallah et al., 2020; Zhou et al., 2019; Rajeswaran et al., 2019; Collins et al., 2020b). Inspired by MAML, a subseqent line of gradient-based meta-learning algorithms have been developed and widely used (Nichol et al., 2018; Al-Shedivat et al., 2017; Jerfel et al., 2018). Much follow-up work focused on the online setting, with accompanying regret bounds (Denevi et al., 2018; Finn et al., 2019; Khodak et al., 2019; Balcan et al., 2015; Alquier et al., 2017; Bullins et al., 2019; Pentina & Lampert, 2014).

The statistical analysis of meta-learning traces back to Baxter (2000); Maurer & Jaakkola (2005), which focused on inductive bias learning. Following the same setting, Amit & Meir (2018); Konobeev et al. (2020); Maurer et al. (2016); Pentina & Lampert (2014) fix a meta-distribution for source task sampling, and measure the average generalization error/gap over the meta distribution. Another line of work connects the target performance to source using distance measures between distributions (Ben-David & Borbely, 2008; Ben-David et al., 2010; Mohri & Medina, 2012). Finally,

another series of works studied the benefits of using additional "side information" provided with a task for specializing the parameters of an inner algorithm (Denevi et al., 2020, 2021).

The hardness of meta-learning has also been investigated under various settings. Recent work has studied meta-learning performance in worst-case settings (Collins et al., 2020b; Hanneke & Kpotufe, 2020a,b; Kpotufe & Martinet, 2018; Lucas et al., 2020). Hanneke & Kpotufe (2020a) provide a no-free-lunch result with a problem-independent minimax lower bound, while Konobeev et al. (2020) also provide problem-dependent lower bound on a simple linear setting.

## 2   General Setting

### 2.1   Notation

Let $[n] \coloneqq \{1, \ldots, n\}$. We denote the vector $L_2$-norm as $\|\cdot\|_2$, and the matrix Frobenius norm as $\|\cdot\|_F$. Additionally, $\langle \cdot, \cdot \rangle$ can denote either the Euclidean inner product or the Frobenius inner product between matrices.

For a matrix $A$, we let $\sigma_i(A)$ denote its $i^{\text{th}}$ largest singular value. Additionally, for positive semidefinite $A$, we write $\lambda_{\max}(A)$ and $\lambda_{\min}(A)$ for its largest and smallest eigenvalues, and $A^{1/2}$ for its principal square root. We write $P_A$ for the projection onto the column span of $A$, denoted $\operatorname{Col} A$, and $P_A^{\perp} \coloneqq I - P_A$ for the projection onto its complement.

We use standard $O, \Theta$, and $\Omega$ notation to denote orders of growth. We also use $a \lesssim b$ or $a \ll b$ to indicate that $a = O(b)$. Finally, we write $a \asymp b$ for $a = \Theta(b)$.

### 2.2   Problem Setting

**Preliminaries**. Fix an input space $\mathcal{X} \subseteq \mathbb{R}^d$ and label space $\mathcal{Y} \subseteq \mathbb{R}$. We assume that the learner has access to $T$ source tasks. Each task $t \in [T]$ is associated with a distribution $\mu_t$ over the set $\mathcal{X} \times \mathcal{Y}$ of input-label pairs. From each $\mu_t$, the learner observes $n_{\mathrm{S}}$ i.i.d. samples $\{(x_{i,t}, y_{i,t}) \mid i \in [n_{\mathrm{S}}]\}$, which we collect into a matrix $X_t \in \mathbb{R}^{n_{\mathrm{S}} \times d}$ and a vector $y_t \in \mathbb{R}^{n_{\mathrm{S}}}$. Finally, we measure learner performance using the loss function $\ell : \mathbb{R} \times \mathcal{Y} \to \mathbb{R}$.

We aim to find common structure among the source tasks that could be leveraged for future tasks. In this work, such common structure is defined using *representations*, mappings $\phi : \mathcal{X} \to \mathcal{Z}$ from the input space to a latent space $\mathcal{Z} \subseteq \mathbb{R}^k$. These representations lie in some function class parameterized by a normed space. However, unlike prior work, we *do not* assume that all tasks use a single representation. Instead, there is a fixed $\theta_0$ such that the predictor for task $t$ can be written as $x \mapsto \langle w_t, \phi_{\theta_t}(x) \rangle$, with $\theta_t$ lying near $\theta_0$ (that is, $\|\theta_t - \theta_0\|$ is small).

**ADAPTREP Procedure**. Following the discussion above, consider the objective

$$\min_{\theta_0} \min_{\substack{\theta_t, w_t \\ \|\theta_t - \theta_0\| \leq \delta_0}} \frac{1}{n_{\mathrm{S}} T} \sum_{t=1}^{T} \sum_{i=1}^{n_{\mathrm{S}}} \ell(\langle w_t, \phi_{\theta_t}(x_{i,t}) \rangle, y_{i,t}) \tag{1}$$

for some fixed $\delta_0 > 0$, which will be the main focus of this work. We refer to (1) as ADAPTREP, as it finds an initialization $\phi_{\theta_0}$ for which every task has a good representation nearby, ensuring ease of fine-tuning. (1) can be viewed as a constrained form of algorithms found in the literature such as iMAML (Rajeswaran et al., 2019) and Meta-MinibatchProx (Zhou et al., 2019). However, we do not use train-validation splits, as is widespread in practice. This is motivated by results in Bai et al. (2020), which show that data splitting may be undesirable, assuming realizability. Furthermore, empirical evaluations have demonstrated successes despite the lack of such splits (Zhou et al., 2019).

**FROZENREP Procedure**. As a baseline, we will also be studying the ubiquitous "frozen representation" objective, which is analyzed in detail in Du et al. (2020); Tripuraneni et al. (2020b). Formally, this objective can be written as follows:

$$\min_{\theta_0} \min_{w_t} \frac{1}{n_{\mathrm{S}} T} \sum_{t=1}^{T} \sum_{i=1}^{n_{\mathrm{S}}} \ell(\langle w_t, \phi_{\theta_0}(x_{i,t}) \rangle, y_{i,t}). \tag{2}$$

Since $\phi_{\theta_0}$ is frozen for all source tasks, we refer to the method above as FROZENREP. In Sections 3.4 and G, we will prove that unlike ADAPTREP, there are cases where FROZENREP is unable to leverage the latent structure in the presence of misspecifications.

**Learning Target Tasks**. Let $\theta_0$ be a solution to (1) or (2). The learner obtains $n_T$ i.i.d. samples $\{(x_i, y_i)\}$ from a new task with distribution $\mu$, which we collect into $X \in \mathbb{R}^{n_T \times d}$ and $y \in \mathbb{R}^{n_T}$ as before. We adapt to the target task using $\theta_0$ by solving

$$\min_{\substack{\theta, w \\ \|\theta - \theta_0\| \leq \delta}} \frac{1}{n_T} \sum_{i=1}^{n_T} \ell(\langle w, \phi_\theta(x_i) \rangle, y_i). \tag{3}$$

for some $\delta \geq 0$. This learned predictor is evaluated via its population loss over $\mu$, which is given by $\mathbb{E}_{(x,y) \sim \mu}[\ell(\langle w, \phi_\theta(x) \rangle, y)]$. We focus on the few-shot learning setting for the target task, where $n_T$ is small, and a learner needs to effectively use the source tasks to sufficiently learn the target task.

**Statistical Assumptions and Optimization Oracles**. For ADAPTREP to be sensible, we need to ensure that a desirable initialization exists. Accordingly, we assume that for some $\theta_0^*$, every task $t$ is associated with $(\theta_t^*, w_t^*)$ where $\|\theta_t^* - \theta_0^*\| \leq \delta_0$, so that the associated distribution $\mu_t$ is given by

$$x \sim p, \quad y \mid x \sim q(\cdot \mid \langle w_t^*, \phi_{\theta_t^*}(x) \rangle).$$

Here, $p$ is an input distribution (same for all tasks), while $q$ models label noise. Specifically, in regression, we set $\mathbb{E}[y \mid x] = \langle w_t^*, \phi_{\theta_t^*}(x) \rangle$. With an appropriate choice of loss function $\ell$, we can guarantee that the optimal predictor under the population loss is $x \mapsto \langle w_t^*, \phi_{\theta_t^*}(x) \rangle$.

Throughout the paper, we assume access to an oracle for solving (1) or (2), similar to Du et al. (2020); Tripuraneni et al. (2020b). For detailed analyses of source training optimization, we refer the reader to Ji et al. (2020); Wang et al. (2020). On the other hand, representation fine-tuning introduces nonconvexity during target time not present in prior work, where one simply optimizes a final linear layer (a convex problem). Thus, our bounds explicitly incorporate optimization performance on (3). We focus on using projected gradient descent (PGD), which applies to a wide variety of settings, under certain loss landscape assumptions. These standalone results are also provided in Section H.

# 3 ADAPTREP in the Linear Setting

We first examine ADAPTREP in the linear setting to illustrate key ideas. Here, representations are linear transformations $\mathbb{R}^d \to \mathbb{R}^k$ for $d > k$ so that $\phi_B(x) = B^\top x$, with distances measured by the Frobenius norm. In this setting, we provide a performance bound for ADAPTREP, and then exhibit a specific instance for which FROZENREP fails, establishing our claimed sample complexity gap.

## 3.1 Statistical Assumptions

**Data Sampling**. We specialize the statistical assumptions of Section 2.2. Assume that the input distribution $p$ is zero-mean and has covariance $\Sigma$. Let $\kappa = \lambda_{\max}(\Sigma)/\lambda_{\min}(\Sigma)$ be the condition number of $\Sigma$. As in Du et al. (2020), we impose the following tail condition on $p$:

**Assumption 3.1.** *There exists $\rho > 0$ such that if $x \sim p$, then $\Sigma^{-1/2}x$ is $\rho^2$-sub-Gaussian[1].*

This assumption guarantees probabilistic tail bounds for our proofs, and can be replaced with other suitable tail conditions. Finally, we define $q(\cdot \mid \mu) \sim \mathcal{N}(\mu, \sigma^2)$ for a fixed $\sigma > 0$.

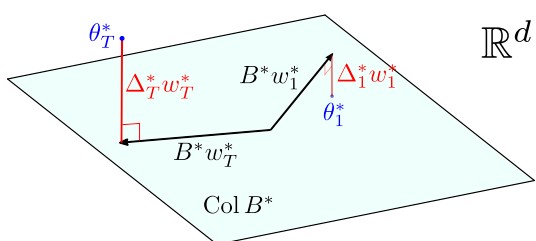

Figure 1: Illustrating the geometry of task parameterization in the linear representation setting.

**Parameterizing the Tasks**. Let $B^*$ be the initial representation, and $\Delta_t^*$ be the task-specific change for task $t$, where $\|\Delta_t^*\|_F \leq \delta_0$. Furthermore, let $w_t^* \in \mathbb{R}^k$ be the task-specific weights. The predictor for task $t$ is then given by $x \mapsto \langle x, (B^* + \Delta_t^*)w_t^* \rangle$. We define $\theta_t^* = (B^* + \Delta_t^*)w_t^*$, and combine $(w_t^*)$ into a matrix $W^* = [w_1^*, \ldots, w_T^*] \in \mathbb{R}^{k \times T}$. We illustrate the parameterization in Figure 1.

---

[1]A zero-mean random vector $v$ is $\rho^2$-sub-Gaussian if for any fixed $u$, $\mathbb{E}[\exp(\lambda v^\top u)] \leq \lambda^2 \|u\|_2^2 \rho^2/2$.

**Assumption 3.2** (Source task diversity)**.** *For any $t \in [T]$, $\|w_t^*\|_2 = \Theta(1)$, and $\sigma_k^2(W^*) = \Omega(T/k)$.*

Since $\sum_{i \in [k]} \sigma_i^2(W^*) = \|W^*\|_F^2 = \Theta(T)$ by Assumption 3.2, the bound on $\sigma_k^2(W^*)$ implies that the weights covers all directions in $\mathbb{R}^k$ roughly evenly. This condition is satisfied with high probability when the $w_t^*$ are sampled from a sub-Gaussian distribution with well-conditioned covariance.

Note that $\theta_t^* = B^* w_t^* + \delta_t^*$, where $\|\delta_t^*\|_2 = \|\Delta_t^* w_t^*\|_2 = O(\delta_0)$ by Assumption 3.2. Thus, the predictor weights lie *near the rank-$k$ subspace determined by $B^*$*, as illustrated by Figure 1. As a convention since the parameterization is not unique, we require $(B^*)^\top \Sigma \delta_t^* = 0$ for any $t \in [T]$. This is without loss of generality, as we can always redefine relevant quantities as needed.

Finally, we evaluate the learner on a target task $\theta^* := B^* w^* + \delta^*$ for some $w^*$ and $\|\delta^*\|_2 \le \delta_0$.

## 3.2 Training Procedure

**Source training**. We consider the following regularized form of (1):

$$\min_B \min_{\Delta_t, w_t} \frac{1}{2 n_{\mathrm{S}} T} \sum_{t=1}^T \|y_t - X_t (B + \Delta_t) w_t\|_2^2 + \frac{\lambda}{2} \|\Delta_t\|_F^2 + \frac{\gamma}{2} \|w_t\|_2^2. \tag{4}$$

In Section B, we show that the regularization is equivalent to regularizing $\sqrt{\lambda \gamma} \|\Delta_t w_t\|_2$, consistent with the intuition that $\Delta_t^* w_t^*$ has small norm. This additional regularization is necessary, since (1) only controls the norm of $\Delta_t$, which is insufficient for controlling $\Delta_t w_t$.

**Target training**. Let $B_0$ be the output of (4) after orthonormalizing. We adapt to the target task via

$$\mathcal{L}_\beta(\Delta, w) = \frac{1}{2n} \|y - \beta X (A_{B_0} + \Delta)(w_0 + w)\|_2^2, \tag{5}$$

where $A_{B_0} := [B_0 \; B_0] \in \mathbb{R}^{d \times 2k}$ and $w_0 = [u, -u]$ for a fixed unit-norm vector $u \in \mathbb{R}^k$. This corresponds to training a predictor of the form $x \mapsto \langle x, (A_{B_0} + \Delta)(w_0 + w) \rangle$. We optimize (5) by performing $T_{\mathrm{PGD}}$ steps of PGD with stepsize $\eta$ on (5) with

$$\mathcal{C}_\beta := \{ (\Delta, w) \mid \|\Delta\|_F \le c_1 / \beta, \|w\|_2 \le c_2 / \beta \}$$

as the feasible set, where we explicitly define $c_1$ and $c_2$ in Section B.

To understand the procedure, note that the predictor has parameter $\theta = \beta(A_{B_0} w + \Delta w_0 + \Delta w)$, since $A_{B_0} w_0 = 0$.[2] Within $\mathcal{C}_\beta$, the first two terms have $O(1)$ norm, while $\|\Delta w\|_2 = O(1/\beta)$. Thus, for large $\beta$, the cross term $\Delta w$ is a negligible perturbation, and the predictor is approximately linear in $(\Delta, w)$. Consequently, $\mathcal{L}_\beta$ is approximately convex in $(\Delta, w)$ within $\mathcal{C}_\beta$, allowing PGD to find a near-optimal solution in the constraint set.

## 3.3 Performance Bound

Now, we provide a performance bound on the performance of the algorithm proposed in Section 3.2. We define the following rates[3] of interest, where $R_\Sigma(w^*, \delta_0) > 0$:

$$r_{\mathrm{S}}(n_{\mathrm{S}}, T) := \frac{\sigma^2 k d}{n_{\mathrm{S}} T} + \sigma \delta_0 \|\Sigma\|_2^{1/2} \sqrt{\frac{kd}{n_{\mathrm{S}} T}} + \frac{\sigma^2 k}{n_{\mathrm{S}}} + \frac{\sigma \delta_0}{\sqrt{n_{\mathrm{S}}}} \sqrt{\operatorname{tr} \Sigma}$$

$$r_{\mathrm{T}}(n_{\mathrm{T}}) := \frac{\sigma^2 k}{n_{\mathrm{T}}} + \frac{\sigma \delta_0}{\sqrt{n_{\mathrm{T}}}} \sqrt{\operatorname{tr} \Sigma} + \min\left[ k r_{\mathrm{S}}(n_{\mathrm{S}}, T), \; R_\Sigma(w^*, \delta_0) \sqrt{k r_{\mathrm{S}}(n_{\mathrm{S}}, T)} \frac{\sigma}{\sqrt{n_{\mathrm{T}}}} \sqrt{\operatorname{tr} \Sigma} \right]$$

**Theorem 3.1** (Performance guarantee, linear representations)**.** *Assume that Assumptions 3.1 and 3.2 hold, $n_{\mathrm{S}} \gg \rho^4(d + \log(T/\delta))$, and $n_{\mathrm{T}} \gg \rho^4(d + \log(1/\delta))$. Then there are $(\lambda, \gamma, \beta, T_{\mathrm{PGD}}, \eta, c_1, c_2)$ (specified in Section B) such that the training procedure in Section 3.2, with high probability, finds $\theta$ achieving excess risk bounded as*

$$\mathbb{E}\left[ (x^\top \theta^* - x^\top \theta)^2 \right] \lesssim \min\left( r_{\mathrm{T}}(n_{\mathrm{T}}), \frac{\sigma^2 d}{n_{\mathrm{T}}} \right).$$

---

[2]This "symmetrization" via $A_{B_0}$ is commonly used in the neural tangent kernel literature (Chizat et al., 2019; Zhang et al., 2020), allowing us to set $f_{\theta_0}(x) \equiv 0$ while ensuring $A_{B_0} w \in \operatorname{Col} B$ for any $w$.

[3]Log factors and non-dominant terms suppressed for clarity. Full rates are presented in the appendix.

We prove this result in Section B. Nevertheless, we give an interpretation of $r_S$ and $r_T$ below.

**Source Training.** The rate $r_S$ represents the performance of the learner during source training. Despite misspecifications, the algorithm is able to pool the aggregate $n_S T$ samples to learn a common ($kd$-dimensional) initialization, as shown by the first two terms. The last two terms represent the rate achieved on each task *upon inferring the initialization*, and therefore decays only with $n_S$. The first is the cost of learning $w_t^*$, while the second is the cost of learning $\delta_t^*$.

**Target Training.** The rate $r_T$ represents the performance of the learner *with fine-tuning*. The first two terms and their interpretations are analogous to that of the last two terms in $r_S$. On the other hand, the minimum indicates two ways to handle the representation error from source training:

*Ignore source error.* In the most data-starved regime, we ignore the finite-sample source error, restricting fine-tuning to only learn $\delta^*$. Thus, we incur an irreducible error in the form of $kr_S(n_S, T)^4$.

*Fix source error.* Alternatively, with more data, we can use fine-tuning to *fix the source representation error*. This additional complexity is captured by the second argument, indicating fine-tuning to a vector with norm $R_\Sigma(w^*, \delta_0)\sqrt{kr_S(n_S, T)}$. Note that the additional norm shrinks to 0 as $n_S \to \infty$.

Finally, by ignoring the representation $B_0$ completely, we can obtain the alternative trivial $\sigma^2 d/n_T$ rate, which matches the minimax lower bound for standard linear regression (see e.g., Duchi (2016)).

## 3.4 A Hard Case for FROZENREP

In this section, we present a sample complexity separation between ADAPTREP and FROZENREP in the linear setting. *We provide the formal construction and an experimental verification of the gap in Section C.* Nevertheless, we provide the following informal result:

**Theorem 3.2** (FROZENREP-ADAPTREP Sample Complexity Separation). *Assume $k = \Theta(1) \ll d$. There exists a family of task distributions $\mathcal{T}$ satisfying the conditions of Section 3.1, such that with high probability over the draw of $n_T \gtrsim d$ target samples, we have the minimax bound*

$$\min_{\hat{w}, \hat{\delta}} \max_{\substack{\tau \in \mathcal{T} \\ \theta^* \in \text{supp } \tau}} \mathbb{E}\left[\frac{1}{n_T}\left\|X(\theta^* - \bar{B}\hat{w} - \hat{\delta})\right\|_2^2\right] \gtrsim \frac{\sigma^2 d}{n_T}.$$

FROZENREP *is provided with infinitely many source tasks and per-task samples from $\tau$ to learn $\bar{B}$, and $n_T$ samples from $\theta^*$ to learn $(\hat{w}, \hat{\delta})$. In contrast, by specializing Theorem 3.1 for the same family,* ADAPTREP *(with only finitely many source tasks and per-task samples) achieves excess risk bounded as*

$$\mathbb{E}\left[(x^\top \theta^* - x^\top \hat{\theta})^2\right] \lesssim \min\left(\frac{\sigma}{\sqrt{n_T}}, \frac{\sigma^2 d}{n_T}\right).$$

**Interpreting the minimax result**. Observe that the rate is achievable by performing standard linear regression directly on the target samples, with no other information. Thus, FROZENREP fails to capture the shared structure from source tasks, in the worst case. Furthermore, in the high-dimensional setting when $n_T \asymp d$, there exists a strict separation between ADAPTREP and FROZENREP that widens as $n_T \to \infty$.

The minimax bound applies regardless of the target fine-tuning procedure in use, including those used in practice, *e.g.* iMAML, MetaOptNet (Lee et al., 2019), and R2D2 (Bertinetto et al., 2019). As FROZENREP is provided with infinitely many source tasks and samples, this failure is thus entirely due to the representation learning algorithm. Altogether, we have established a case where incorporating representation fine-tuning is *provably sufficient* for handling misspecifications, unlike ubiquitous "frozen representation" objectives.

### 3.4.1 Intuition

In this section, we provide a brief intuition behind our construction. Recall that in the linear setting, FROZENREP uses the following objective for obtaining a representation:

$$\hat{B} = \underset{B}{\arg\min} \min_{w_t} \frac{1}{2n_S T} \sum_{t \in [T]} \|y_t - X_t B w_t\|_2^2 \xrightarrow{n_S \to \infty} \frac{1}{2T} \sum_{t \in [T]} \|\theta_t^* - B w_t^*\|_\Sigma^2. \quad (6)$$

---

[4]For *average* target performance, $kr_S(n_S, T)$ can be replaced by $r_S(n_S, T)$; see Section F; Du et al. (2020).

By optimizing with respect to the $\Sigma$-norm, FROZENREP seeks a $k$-dimensional subspace that captures as much of the predictive signal across all source tasks. However, in general, the $\Sigma$-norm is not aligned with the parameter norm. In other words, a small $\|\theta - Bw\|_\Sigma$ does not imply that $\|\theta - Bw\|$ is small, and thus it may be the case that $\theta - Bw$ is difficult to learn during fine-tuning.

A prototypical example of this difference is illustrated by Figure 2, where $k = 1$. The dashed red vector has small parameter norm, and is thus easily learnable by fine-tuning (as demonstrated by norm-based generalization bounds). Thus, a meta-learner should opt to learn the space captured by the solid green vector. However, if the $\Sigma$ is structured so that $\|\Delta_t^* w_t^*\|_\Sigma \gg \|B^* w_t^*\|_\Sigma$, then FROZENREP will learn the space spanned by the dashed red vector instead. By appropriately tuning the ratio of the lengths of the two vectors, we obtain the desired result.

We provide a formal construction in Section C.1.1, including the extension to general $k$.

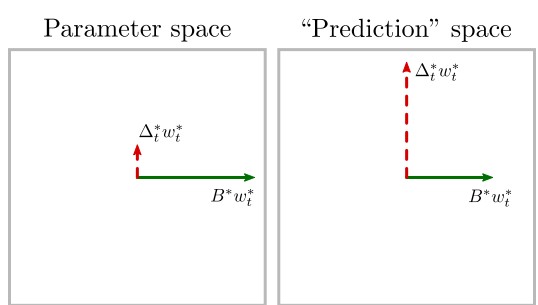

Figure 2: Prototypical hard case. While $\Delta_t^* w_t^*$ is smaller (left figure), FROZENREP learns $\Delta_t^* w_t^*$ since it has a larger predictive signal (right figure).

# 4 ADAPTREP in the Nonlinear Setting

We now describe a framework for analyzing fine-tuning in general function classes. To simplify notation, we modify the setting in Section 2.2 so that both the representation $\phi$ and predictor weights $w$ are captured by a parameter $\theta \in \Theta$, with corresponding predictor $g_\theta$.

**Loss Notation**. We denote the population and excess risk of predictor $g$ with respect to the true predictor $g^*$ as

$$\mathcal{L}_\infty(g, g^*) \coloneqq \mathbb{E}_{x \sim p}\left[\mathbb{E}_{y \sim q(\cdot | g^*(x))}\left[\ell(g(x), y)\right]\right]$$
$$\mathcal{L}_\infty^{\mathrm{ex}}(g, g^*) \coloneqq \mathcal{L}_\infty(g, g^*) - \mathcal{L}_\infty(g^*, g^*)$$

respectively. Additionally, we let $\mathcal{L}(g, g^*)$ denote the corresponding finite-sample quantity.[5]

**Feasible Predictors**. For an initialization $\theta$ and a fine-tuning set $\mathcal{C} \subseteq \Theta$, we define the set

$$\mathcal{A}_\mathcal{C}(\theta) \coloneqq \{g_{\theta'} \mid \theta' - \theta \in \mathcal{C}\}.$$

Intuitively, this set contains all feasible predictors given an initialization $\theta$ and feasible fine-tuning set $\mathcal{C}$. For convenience, we also define the set

$$[\mathcal{A}_\mathcal{C}(\theta)]^{\otimes T} \coloneqq \left\{(x_t)_{t \in [T]} \mapsto (g_t(x_t))_{t \in [T]} \mid g_1, \ldots, g_T \in \mathcal{A}_\mathcal{C}(\theta)\right\},$$

which is a collection of functions mapping $\mathcal{X}^T \to \mathcal{Y}^T$. This set represents all feasible $T$-tuples of predictors, for a given initialization $\theta$ and fine-tuning set $\mathcal{C}$.

## 4.1 Training Procedure

**Source Training**. Fix a set of possible initializations $\Theta_0 \subseteq \Theta$, and a fine-tuning set $\mathcal{C}_S$. With $n_S$ samples for each task (with optimal predictor $g_t^*$), we consider the objective

$$\theta_0 \in \underset{\theta \in \Theta_0}{\operatorname{argmin}} \frac{1}{T} \sum_{t=1}^T \min_{g_t \in \mathcal{A}_{\mathcal{C}_S}(\theta)} \mathcal{L}(g_t, g_t^*). \tag{7}$$

Note that this corresponds to the ADAPTREP objective in (1) using the combined parameter $\theta$. We illustrate the objective in Figure 3

**Target Training**. Fix a fine-tuning set $\mathcal{C}_T$ (not necessarily $\mathcal{C}_S$). Given $n_T$ samples from a new task (with true predictor $g^*$), we run PGD on $L(\delta) = \mathcal{L}(g_{\theta_0 + \delta}, g_t^*)$ with feasible set $\mathcal{C}_T$. In this process, PGD is run with $T_{\mathrm{PGD}}$ timesteps and step size $\eta$.

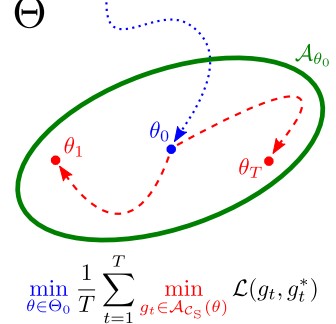

$$\min_{\theta \in \Theta_0} \frac{1}{T} \sum_{t=1}^T \min_{g_t \in \mathcal{A}_{\mathcal{C}_S}(\theta)} \mathcal{L}(g_t, g_t^*)$$

Figure 3: An illustration of source training.

---

[5]Note that we have omitted the samples from the notation for brevity.

## 4.2 Assumptions

We rephrase the statistical assumption from Section 2.2 in terms of feasible sets $\mathcal{A}_{\mathcal{C}}(\theta)$ as follows:

**Assumption 4.1.** *There exists $\theta_0^*$ such that for every $t \in [T]$, $g_t^* \in \mathcal{A}_{\mathcal{C}_{\mathrm{S}}}(\theta_0)$.*

We illustrate Assumption 4.1 in Figure 4. We also impose the following standard regularity conditions on the loss function $\ell$:

**Assumption 4.2.** *For any $y \in \mathcal{Y}$, $\ell(\cdot, y)$ is 1-Lipschitz[6] and convex, and $|\ell(0, y)| \leq B$.*

**Task Diversity**. To ensure transfer from source to target, we impose the following condition, a specific instance of which was proposed by Du et al. (2020) in the linear setting, and proposed by Tripuraneni et al. (2020b) for general settings:

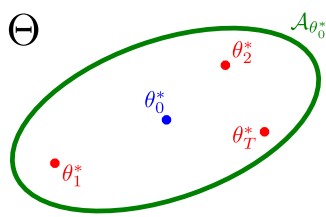

Figure 4: Assumption 4.1. If $\mathcal{A}_{\mathcal{C}}(\theta)$ has low complexity, then fine-tuning quickly finds good task-specific parameters for all tasks.

**Assumption 4.3** (Source tasks are $(\nu, \varepsilon)$-diverse). *There exists constants $(\nu, \varepsilon)$ such that if $\rho$ is the distribution of target tasks, then for any $\theta \in \Theta_0$,*

$$\mathbb{E}_{g^* \sim \rho} \left[ \inf_{g \in \mathcal{A}_{\mathcal{C}_{\mathrm{T}}}(\theta)} \mathcal{L}_{\infty}^{\mathrm{ex}}(g, g^*) \right] \leq \frac{1}{\nu} \left[ \frac{1}{T} \sum_{t=1}^{T} \inf_{g \in \mathcal{A}_{\mathcal{C}_{\mathrm{S}}}(\theta)} \mathcal{L}_{\infty}^{\mathrm{ex}}(g, g_t^*) \right] + \varepsilon.$$

The $(\nu, \varepsilon)$-diversity assumption connects the average source task performance (right) to the average target performance (left). Thus, it ensures that optimizing (7) results in controlled target performance. We weakened the condition in Tripuraneni et al. (2020b) to bound average target performance rather than worst-case, as is more suitable for higher-dimensional settings.

**Target Loss Landscape**. Finally, we impose conditions ensuring that PGD finds a near-optimal solution in $\mathcal{C}_{\mathrm{T}}$. We remark that these assumptions are specific to our choice of fine-tuning algorithm.

**Assumption 4.4** (Approximate linearity in fine-tuning). *Let $x_1, \dots, x_{n_{\mathrm{T}}}$ be the set of target inputs. Then, there exists $\beta, L$ such that*

$$\sup_{\delta \in \mathcal{C}_T} \frac{1}{n_{\mathrm{T}}} \sum_{i=1}^{n_{\mathrm{T}}} \left\| \nabla_{\theta}^2 g_{\theta+\delta}(x_i) \right\|_2^2 \leq \beta^2 \quad \text{and} \quad \sup_{\theta \in \Theta_0} \frac{1}{n_{\mathrm{T}}} \sum_{i=1}^{n_{\mathrm{T}}} \left\| \nabla_{\theta} g_{\theta}(x_i) \right\|_2^2 \leq L^2.$$

**Assumption 4.5.** $\sup_{\delta \in \mathcal{C}_{\mathrm{T}}} \|\delta\|_2 \leq R$ *for some $R$.*

## 4.3 Performance Bound

Recall that for a set $\mathcal{H}$ of functions $\mathbb{R}^d \to \mathbb{R}^k$ on $n$ samples, its Rademacher complexity $\mathcal{R}_n(\mathcal{H})$ on $n$ samples is given by

$$\mathcal{R}_n(\mathcal{H}) \coloneqq \mathbb{E}_{\varepsilon, X} \left[ \frac{1}{n} \left| \sup_{h \in \mathcal{H}} \sum_{i=1}^{n} \sum_{j=1}^{k} \varepsilon_{ij} h_j(x_i) \right| \right],$$

where $(\varepsilon_{ij})$ are i.i.d. Rademacher random variables and $(x_i)$ are i.i.d. samples from some (preset) distribution. We now proceed with our performance guarantee.

**Theorem 4.1** (General Performance Bound). *Assume that all assumptions in Section 4.2 hold. Let $(\theta_t)$ be the set of iterates generated by PGD following the procedure in Section 4.1 (step size $\eta$ specified in Section D). Then, with probability at least $1 - \delta$ over the random draw of samples,*

$$\mathbb{E}_{g^* \sim \rho} \left[ \min_t \mathcal{L}_{\infty}^{\mathrm{ex}}(g_{\theta_t}, g^*) \right] \lesssim \underbrace{\beta R^2 + R\sqrt{\frac{L^2 + \beta^2 R^2}{T_{\mathrm{PGD}}}}}_{\varepsilon_{\mathrm{OPT}}} + \underbrace{\frac{1}{\delta} \sup_{\theta \in \Theta_0} \mathcal{R}_{n_{\mathrm{T}}} \left[ \mathcal{A}_{\mathcal{C}_{\mathrm{T}}}(\theta) \right] + \frac{B}{\delta \sqrt{n_{\mathrm{T}}}}}_{\varepsilon_{\mathrm{EST}}}$$

$$+ \underbrace{\frac{1}{\nu} \left\{ \frac{1}{\delta T} \mathcal{R}_{n_{\mathrm{S}}} \left[ \bigcup_{\theta \in \Theta_0} [\mathcal{A}_{\mathcal{C}_{\mathrm{S}}}(\theta)]^{\otimes T} \right] + \frac{B}{\delta \sqrt{n_{\mathrm{S}} T}} \right\}}_{\varepsilon_{\mathrm{REPR}}} + \varepsilon.$$

---

[6]This is not restrictive as one can simply rescale the loss, and is assumed for simplicity of presentation.

*Note that the $\mathcal{R}_{n_\mathrm{T}}$ complexity term samples from $p$. Meanwhile, the $\mathcal{R}_{n_\mathrm{S}}$ complexity term samples from $p^{\otimes T}$, which concatenates $T$ i.i.d. samples from $p$ (one for each task) for every draw.*

We prove Theorem 4.1 in Section D. The Rademacher complexity terms decay in most settings as

$$\frac{1}{T}\mathcal{R}_{n_\mathrm{S}}\left[\bigcup_{\theta\in\Theta_0}[\mathcal{A}_{\mathcal{C}_\mathrm{S}}(\theta)]^{\otimes T}\right] = O\left(\frac{C(\Theta_0)}{\sqrt{n_\mathrm{S}T}} + \frac{\mathrm{diam}\,\mathcal{C}_\mathrm{S}}{\sqrt{n_\mathrm{S}}}\right)$$

$$\sup_{\theta\in\Theta_0}\mathcal{R}_{n_\mathrm{T}}[\mathcal{A}_{\mathcal{C}_\mathrm{T}}(\theta)] = O\left(\frac{\mathrm{diam}\,\mathcal{C}_\mathrm{T}}{\sqrt{n_\mathrm{T}}}\right),$$

where $C(\Theta_0)$ measures the complexity of $\Theta_0$, and $\mathrm{diam}\,\mathcal{C}_\mathrm{S}$ and $\mathrm{diam}\,\mathcal{C}_\mathrm{T}$ measures the size of the fine-tuning sets $\mathcal{C}_\mathrm{S}$ and $\mathcal{C}_\mathrm{T}$.

**Understanding the Bound**. We briefly outline out how the assumptions in Section 4.2 contribute to the final bound. Let $\theta_{\mathrm{OPT}}$ denote the best solution found by PGD, and define the parameters

$$\theta_{\mathrm{ERM}} \coloneqq \operatorname*{argmin}_{\theta\in\mathcal{A}_{\mathcal{C}_\mathrm{T}}(\theta_0)} \mathcal{L}(g_\theta, g^*) \quad \text{and} \quad \bar\theta \coloneqq \operatorname*{argmin}_{\theta\in\mathcal{A}_{\mathcal{C}_\mathrm{T}}(\theta_0)} \mathcal{L}_\infty(g_\theta, g^*).$$

That is, $\theta_{\mathrm{ERM}}$ is the minimizer of the empirical risk while $\bar\theta$ is the minimizer of the population risk, *both within $\mathcal{A}_{\mathcal{C}_\mathrm{T}}(\theta_0)$.*

**Optimization error** ($\varepsilon_{\mathrm{OPT}}$). The difference in performance between $\theta_{\mathrm{OPT}}$ and $\theta_{\mathrm{ERM}}$, *i.e.* the error due to PGD, is controlled by the assumptions on the target loss landscape.

**Estimation error** ($\varepsilon_{\mathrm{EST}}$). By uniform convergence, $\theta_{\mathrm{ERM}}$ performs similarly to $\bar\theta$, with the difference bounded using Rademacher complexity. By extension, $\theta_{\mathrm{OPT}}$ performs similarly to $\bar\theta$.

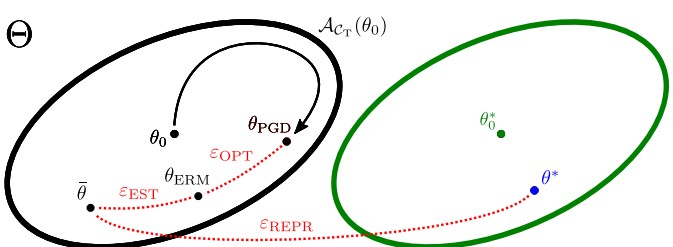

Figure 5: Illustrating the excess risk bound in Theorem 4.1.

**Representation/approximation error** ($\varepsilon_{\mathrm{REPR}}$). The average best case excess risk (i.e., the excess risk of $\bar\theta$) is bounded via the performance of the initialization $\theta_0$ on the source tasks, using $(\nu, \varepsilon)$-diversity. Since (7) is a finite-sample estimate of the desired bound, we again use uniform convergence arguments, which manifest as the second Rademacher complexity term.

We have included an illustration of these three components in Figure 5.

# 5 Case Study: Two-Layer Neural Networks

We now illustrate our framework by instantiating our bounds in concrete settings. We focus on specializing Theorem 4.1 to the two-layer neural networks (this section) and logistic regression (provided in the appendix, Section F). For this section, we fix an activation function $\sigma : \mathbb{R} \to \mathbb{R}$.

**Assumption 5.1.** *For any $t \in [-2, 2]$, $|\sigma'(t)| \leq L$ and $|\sigma''(t)| \leq \mu$. Furthermore, $\sigma(0) = 0$.*

Then, for a constant $\beta$ and $\theta = (B, w)$, where $B \in \mathbb{R}^{d\times 2k}$ and $w \in \mathbb{R}^k$, we define the neural network $f_\theta^\beta(x) = \beta w^\top \sigma(B^\top x)$, where $\sigma$ is applied elementwise.

## 5.1 Statistical Assumptions

**Data Sampling**. For all tasks, the inputs are sampled from a 1-norm-bounded distribution $p$. Furthermore, $q(\cdot \mid \mu)$ is generated as $\mu + \eta$ for some $O(1)$-bounded additive noise $\eta$, similar to Tripuraneni et al. (2020b).

**Parameterizing the Tasks**. We fix a representation $B^* \in \mathbb{R}^{d\times 2k}$ and a linear predictor $w_0^* \in \mathbb{R}^{2k}$. In this setting, we consider representation changes lying in a $k$-dimensional subspace, determined

by an orthonormal[7] set of matrices $\Delta_1^*, \ldots, \Delta_k^*$ in $\mathbb{R}^{d \times 2k}$. Then, for each $t \in [T]$, we have two unit-norm vectors $w_t^*, \delta_t^*$ so that task $t$ has parameter

$$\theta_t^* = \left( B^* + \frac{1}{\beta} \sum_{i \in [k]} (\delta_t^*)_i \Delta_i^* \ , \ w_0^* + \frac{1}{\beta} w_t^* \right).$$

We assume a diversity condition on $(w_t^*, \delta_t^*)$ similar in spirit to the linear version in Assumption 3.2, see Assumption E.2.

With appropriate assumptions on the initialization and large enough $\beta$, the source task predictors behave like their linearizations, *i.e.* there exists feature vectors $\phi_{B^*}$ and $\psi_{B^*, w_0^*}$ so that

$$f_{\theta_t^*}(x) \approx \phi_{B^*}(x)^\top w_t^* + \left\langle \psi_{B^*, w_0^*}(x) \ , \ \sum_{i \in [k]} \delta_{t,i}^* \Delta_i^* \right\rangle.$$

We make use of these assumptions throughout for our result, see Assumption E.1 for the formal statement. Note that $\phi_{B^*}$ and $\psi_{B^*, w_0^*}$ correspond to the "activation" and "gradient" features, respectively, that are empirically evaluated by Mu et al. (2020).

## 5.2   Training Procedure

We use the squared error loss to train in this setting. Additionally, we use the training procedure in Section 4.1 with the function class $\{f_\theta^\gamma\}$, with a minor caveat: we set $\gamma = \beta$ during source training, and $\gamma = O(\sqrt{n_{\mathrm{T}}})$ during target training.

With the procedure above, we now define the relevant feasible sets. We set $\Theta_0$ be the set of initializations satisfying Assumption E.1. Finally, we set the constraint sets

$$\mathcal{C}_{\mathrm{S}} = \{(\Delta, w) \mid \|\Delta\|_F, \|w\|_2 \leq 1/\beta\} \quad \text{and} \quad \mathcal{C}_{\mathrm{T}}^\gamma = \left\{ (\Delta, w) \ \middle| \ \|\Delta\|_F^2 + \|w\|_2^2 \leq O(1/\gamma^2) \right\}.$$

## 5.3   Performance Guarantee

With the above assumptions in hand, we proceed with the performance guarantee.

**Theorem 5.1** (Neural net performance bound). *Assume that Assumptions 5.1, E.1 and E.2 hold. Then, if $n_{\mathrm{S}} \geq n_{\mathrm{T}}$, there exists a setting of the training parameters (see Section E) such that with probability at least $1 - \delta$, the iterates $(\theta_t)$ satisfy*

$$\mathbb{E}_{f^* \sim \rho} \left[ \min_t \mathcal{L}_\infty^{\mathrm{ex}}(f_{\theta_t}^\gamma, f^*) \right] \lesssim L^2 \frac{k}{\sqrt{n_{\mathrm{T}}}} + L(L + \mu) \frac{k^{3/2}}{\sqrt{n_{\mathrm{S}} T}} + \left( \frac{\mu + L}{\beta} \right) L\sqrt{k},$$

*where log factors have been suppressed for clarity.*

# 6   Conclusion

We have presented, to the best of our knowledge, the first statistical analysis of fine-tuning-based meta-learning. We demonstrate the success of such algorithms under the assumption of approximately shared representations between available tasks. In contrast, we show that "frozen representation" objectives analyzed by prior work fail under this weaker assumption.

An interesting line of future work is to determine ways to formulate useful shared structure among MDPs, *i.e.* formulate settings for which meta-reinforcement learning succeeds and results in improved regret bounds for downstream tasks.

# 7   Acknowledgements

KC is supported by a National Science Foundation Graduate Research Fellowship, Grant DGE-2039656. QL is supported by NSF #2030859 and the Computing Research Association for the CIFellows Project. JDL acknowledges support of the ARO under MURI Award W911NF-11-1-0304, the Sloan Research Fellowship, NSF CCF 2002272, NSF IIS 2107304, and an ONR Young Investigator Award.

---

[7] Orthonormal with respect to the Frobenius (*i.e.* entrywise) inner product on matrices.

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
