# OpenReview forum: "How Fine-Tuning Allows for Effective Meta-Learning"
_NeurIPS.cc/2021/Conference — NeurIPS 2021 Poster_

### Official Review · Reviewer_dKjD · 2021-07-01

**Rating:** 7
**Confidence:** 2

**Summary:**

The paper analyses and compares two approaches ("AdaptRep", "FrozenRep") to representation-based meta-learning. Such approaches employ representations $\phi_\theta(x)$, defining a predictor via $<w, \phi_\theta(x)>$. Both $\theta$ and $w$ are optimised on the target task, with $\theta$ only being fine-tuned, i.e., constrained to stay close to some initialisation $\theta_0$.

The analysed approaches differ in how the initialisation $\theta_0$ is realised. In both cases, $\theta_0$ is optimised on the source tasks in order to leverage shared structure, with the difference that (1) the AdaptRep approach allows fine-tuning on the source tasks, i.e., finds an initialisation s.t. every source tasks has good representation close to $\theta_0$, while (2) the FrozenRep approach does not consider fine-tuning on the source tasks, i.e., optimises for an initialisation which is used for all source tasks without adaptation.

The authors prove performance bounds of AdaptRep in both the linear and non-linear settings. Furthermore, they provide a worst-case example where FrozenRep is not able to leverage information from the source data (i.e., does not perform better than standard linear regression on the target data), while AdaptRep achieves better sample complexity using source information. Finally, the authors apply their theoretical analysis to the concrete setting of a two-layer neural network as well as to logistic regression.

**Ethical Concerns:**

No ethical concerns.

**Limitations And Societal Impact:**

No potential negative societal impact.

**Main Review:**

Originality: The authors claim to provide the first statistical analysis of "AdaptRep"-like algorithms, while previous work focuses on "FrozenRep"-like approaches. They clearly work out this distinction in their Introduction and Related Work section.

Quality: The problem setting as well as the compared approaches are stated clearly and in a technically sound manner. The authors make sure to clearly state relevant assumptions for their proofs. I cannot judge the soundness of the proofs themselves.

Clarity: The paper is written clearly. I think the authors do a good job in guiding the reader through their paper by stating the main results in simplified form in the introduction and giving intuitive explanations of the results in the main chapters. The mathematical details are quite dense and moved to the appendix. I cannot judge the presentation and clarity of the derivations and proofs.

Significance: I think that a sound theoretical analysis of MAML-like algorithms (beyond FrozenRep-like approaches) is important and therefore I think the results are interesting for the community. However, I am not familiar enough with the state of the art to fully judge the significance of this work.

In summary, I think this work is an interesting contribution and presented in a clear and well-written manner. Therefore, I recommend acceptance. However, this recommendation  is with low confidence, as I am not very familiar with the theoretical analysis of meta-learning algorithms. Therefore, I might adjust my rating after the discussion period.

**Time Spent Reviewing:**

3

---

> ### Author Response · Authors · 2021-08-10
> **Response to Reviewer dKjD**
>
> We thank the reviewer for their review, and we appreciate their comments regarding our technical contributions and the readability of our work. We will work towards adding more discussions regarding our theorems, and we aim to make the work more accessible as a whole (e.g. with examples, intuitive proof sketches like in Section B.1.1, etc.)

---

### Official Review · Reviewer_4qNe · 2021-07-17

**Rating:** 6
**Confidence:** 3

**Summary:**

This paper proposed a theoretical framework for analyzing a type of MAML-like meta-learning algorithms. The algorithm learns an initialization for the representation from a collection of source tasks, with the goal that there is a good representation close to the initialization, as well as an unconstrained predictor, for each of source tasks. Then on a target task, it fine-tunes the representation and trains a predictor altogether. The paper provides risk bounds in linear settings, and points out a separating case where the “frozen representation” approach by Du et al. (2020) cannot achieve the same guarantee. The results are also extended to the general case such as two-layer neural networks.

**Limitations And Societal Impact:**

Yes, in the Checklist.

**Main Review:**

- The paper focused on an important formulation of initialization-based meta-learning algorithms. Different from previous work, this paper assumes source tasks, as well as the target task, share only approximately the same representation. That is, it allows for fine-tuning the representation from a learned initialization. The setting is more practical but less theoretically understood. This paper provides a series of theoretical guarantee for risks in linear and more general settings. The results are novel and may be of interest for a broad audience in meta-learning.
- One concern is the significance of the proposed framework over the frozen representation approach by Du et al. (2020). This paper shows that allowing fine-tuning can achieve strictly better risk bounds than the frozen representation algorithms, but limited to a linear case where tasks are formulated from a shared optimal representation within a small range of perturbation. How does AdaptRep compare to FrozenRep when faced with harder tasks? In particular, how does the general performance bound in Section 4.3 advance over previous work? A detailed discussion would be helpful.
- In practical algorithms like MAML, the whole model including the representation and predictor is often fine-tuned together. Would the proposed framework provide theoretical guarantee for this setting?
- In introduction some quantitative results are shown without giving definitions for involved variables, which come clear only in Section 2. The main body of the paper is sometimes notationally dense. It would improve readability if more discussion is provided.


**Time Spent Reviewing:**

4

---

> ### Author Response · Authors · 2021-08-10
> **Response to Reviewer 4qNe**
>
> We thank the reviewer for their review and appreciate their encouraging comments. We address the reviewer’s comments and concerns in order:
>
> > One concern is the significance of the proposed framework over the frozen representation approach by Du et al. (2020). This paper shows that allowing fine-tuning can achieve strictly better risk bounds than the frozen representation algorithms, but limited to a linear case where tasks are formulated from a shared optimal representation within a small range of perturbation. How does AdaptRep compare to FrozenRep when faced with harder tasks? In particular, how does the general performance bound in Section 4.3 advance over previous work? A detailed discussion would be helpful.
>
> Our appendix includes an extension of the ideas from the linear hard case to the shallow ReLU setting in Section F. Within that setting, we demonstrate an analogous sample complexity separation between AdaptRep and FrozenRep.
>
> Additionally, while the assumptions of our framework may seem superficially similar to that of Du et al. (2020), the techniques required to specialize our framework to a concrete setting requires different arguments of independent interest. In particular, proving a concrete $(\nu, \epsilon)$-diversity assumption requires additional work to successfully decouple the learning of the representation and the task-specific residuals (e.g. see Lemmas A.4, D.4, E.1)
>
> > In practical algorithms like MAML, the whole model including the representation and predictor is often fine-tuned together. Would the proposed framework provide theoretical guarantee for this setting?
>
> All of our results are for the case where the representation and the predictor are fine-tuned together; in our analysis, gradient descent is assumed to be optimizing both the representation and the predictor simultaneously.
>
> > In introduction some quantitative results are shown without giving definitions for involved variables, which come clear only in Section 2. The main body of the paper is sometimes notationally dense. It would improve readability if more discussion is provided.
>
> We will work on improving and streamlining our discussion of our quantitative results throughout the paper. Additionally, we will work on simplifying our notation, and including a notation glossary for convenience.
>
> We hope that the above addresses the reviewer’s concerns, and that the reviewer will increase their score with these clarifications, if they are satisfactory. We will incorporate the discussion points above to improve the clarity of our work, and we would be happy to further elaborate/respond to any other concerns that the reviewer may have during the rolling discussion period!

---

### Official Review · Reviewer_Jbdo · 2021-07-19

**Rating:** 5
**Confidence:** 2

**Summary:**

This paper studies the problem of regret bounds for meta-learning where representations are fine-tuned for individual tasks.

The main contribution of the paper is a theoretical result, showing that (under certain assumptions) learning a fixed representation across all tasks is no better than just using an individual task's data alone. Thus, fixed representations cannot leverage having information from additional tasks. However, the paper also shows that adaptive representations (cases where the representation is allowed to be tuned a little bit for a new task) do not suffer in the same way.

Overall, this provides some theoretical support to recent empirical successes using MAML-like algorithms that fine-tune representations through meta-learning.

**Main Review:**

This paper motivates and studies an important problem. Adding theoretical understanding to when and why meta-learning works is an important problem. This paper focuses on MAML-like algorithms, that is, algorithms that fine tune a representation for a new task. The starting point for the fine-tuning is meta-learned across a distribution of tasks.

My main concerns with the paper have to do with presentation. While the paper is largely well written, I found myself having a hard time picturing or grasping the intuition behind some of the theoretical results. Perhaps including some kind of simple graphics demonstrating the problem setting might help readers quickly grasp the framework and assumptions used to build the theoretical results.

Regarding the main result, I have some questions about the assumptions made about task distributions. Intuitively, I feel like whether or not this theoretical finding is surprising rests on the assumptions made about the task distribution. How similar are tasks drawn from some task family? That is a hard question, it is not clear to me how to even define similarity across tasks. However, in order for meta-learning to be useful, there must be _some_ shared structure or information across tasks. In this paper, the shared structure across tasks is outlined in the section "Statistical Assumptions and Optimization Oracles". If I understand correctly, the paper assumes that:
(1) data points across tasks are drawn from the same distribution (p)
(2) labels for a given task are drawn from a distribution conditioned on a representation near a shared representation.
Basically, this means that there is no model mismatch between the AdaptRep algorithm and the tasks. But there is a mismatch between the FrozenRep algorithm and the underlying data distribution--the modeling assumption of the FrozenRep algorithm (that a single representation is used across tasks) is not valid. As such, I am not sure how surprised I should be by the gap in the regret bounds.

I am also left wondering, are these assumptions valid in practice? For the kinds of tasks where MAML-like algorithms have succeeded, do these assumptions roughly hold? Empirically, can you show in toy settings that the AdaptRep algorithm works when these assumptions are held, but doesn't work when the assumptions are broken?

Overall, I think the paper would be improved with more intuition both on the assumptions behind the method, and on when the assumptions will or won't hold in practice.

**Time Spent Reviewing:**

4

---

> ### Author Response · Authors · 2021-08-10
> **Response to Reviewer Jbdo**
>
> We thank the reviewer for their review. We address the concerns within the review in order:
>
> > My main concerns with the paper have to do with presentation. While the paper is largely well written, I found myself having a hard time picturing or grasping the intuition behind some of the theoretical results. Perhaps including some kind of simple graphics demonstrating the problem setting might help readers quickly grasp the framework and assumptions used to build the theoretical results.
>
> We will work on creating figures demonstrating our assumptions, as well as emphasizing the underlying intuition. For our hard case construction in the linear setting, we also refer the reviewer to Section B.1.1 in the appendix, where we illustrate the main idea behind the construction, and provide intuition as to why FrozenRep can fail. We will try to move the figure and the explanation to the main paper.
>
> > Basically, this means that there is no model mismatch between the AdaptRep algorithm and the tasks. But there is a mismatch between the FrozenRep algorithm and the underlying data distribution--the modeling assumption of the FrozenRep algorithm (that a single representation is used across tasks) is not valid. As such, I am not sure how surprised I should be by the gap in the regret bounds.
>
> We want to reemphasize that FrozenRep and AdaptRep refers to the source-time representation learning method, not the downstream target-time training procedure (i.e. our separation result applies to FrozenRep with any target procedure, _including end-to-end fine-tuning_).
>
> We agree with the reviewer that the modeling mismatch suggests that the AdaptRep-FrozenRep sample complexity gap should indeed exist and is in a way, obvious. However, it is not clear which situations lead to such gaps, especially since FrozenRep can be competitive in practice. Realistically, one expects a certain degree of representation shift in most settings, so it is surprising that FrozenRep can be competitive _despite this model mismatch_.
>
> With the prior discussion in mind, our goal in separating AdaptRep and FrozenRep with our construction is _not to show that there exists a gap_, but to _clarify when such a separation exists_. As our construction shows, FrozenRep attempts to capture the dominant predictive signal across tasks, and thus a sample complexity gap exists when this signal comes from task-specific residuals rather than the shared representation. If the dominant predictive signal comes from the representation, the same proof also demonstrates that FrozenRep learns the same representation as AdaptRep _despite the model mismatch_, and thus is competitive. We illustrate these intuitions in Section B.1.1 in the appendix.
>
> > I am also left wondering, are these assumptions valid in practice? For the kinds of tasks where MAML-like algorithms have succeeded, do these assumptions roughly hold? Empirically, can you show in toy settings that the AdaptRep algorithm works when these assumptions are held, but doesn't work when the assumptions are broken?
>
> We first want to emphasize that our results do apply to end-to-end finetuning, as long as the parameters do not move very far from the pre-trained network relative to the Lipschitz constant of the Jacobian.
>
> Furthermore, Mu et al. [1] evaluated the use of gradients as features in place of fully fine-tuning neural networks. Predictors on gradient features approximate small changes in the network parameters during fine-tuning, corresponding to our assumption of small misspecifications. They demonstrate that using gradient features instead of fine-tuning results in competitive or even better performance on standard benchmark datasets, suggesting that our assumption of small task-specific misspecifications is reasonable.
>
> We will be working on incorporating additional discussion in the final version to clarify our goals and ease the reader into our assumptions/technical results. We hope that the above addresses the reviewer’s concerns, and that the reviewer will increase the score in light of our explanations. We would be more than happy to further discuss these and any other questions the reviewer may have during the rolling discussion period!
>
> [1] Mu, F., Liang, Y., & Li, Y. (2019, September). Gradients as Features for Deep Representation Learning. In International Conference on Learning Representations.

---

> > ### Comment · Reviewer_Jbdo · 2021-08-30
> > **Thank you for your response**
> >
> > Thank you for your response. I am still struggling with intuition, for example, what does it mean for the "dominant predictive signal" to come from "task-specific residuals" rather than the shared representation? Is there a simple picture I should have in mind when trying to parse that statement?

---

> > > ### Author Response · Authors · 2021-08-30
> > > **Clarification**
> > >
> > > We appreciate your response.
> > >
> > > **A simple example.** The statement is easiest to visualize with a linear model $y = x^\top\theta_t + z$, with Gaussian $z$ and Gaussian $x$ with covariance $\Sigma$. In our fine-tuning setting, $\theta_t = (B + \Delta_t)w_t = Bw_t + \Delta_tw_t$, where $B$ is the shared representation and $\Delta_t$ is the task-specific residual. Then, because of linearity, we can rewrite the model as
> > >
> > > $$y = x^\top Bw_t + x^\top \Delta_tw_t + z.$$
> > >
> > > What we mean then by “the dominant signal comes from task-specific residuals rather than the shared representation” is that the _first term contributes less to the magnitude of $y$ than the second_. Formally, assuming $B^\top\Sigma\Delta_tw_t = 0$, we can compute $\mathbb{E}[y^2]$ as
> > >
> > > $$\mathbb{E}[y^2] = \lVert Bw_t\rVert_\Sigma^2 + \lVert \Delta_t w_t\rVert_\Sigma^2 + \text{noise variance}.$$
> > >
> > > and so formally, $\lVert \Delta_tw_t\rVert_\Sigma^2 > \lVert Bw_t\rVert_\Sigma^2$.
> > >
> > > **Visualizing the above.** The formal description suggests a picture of two vectors $Bw_t$ and $\Delta_tw_t$ being compared with respect to the $\Sigma$-norm (which captures their contribution to $y$). _The hard case works since FrozenRep seeks the largest component of $y$ in this $\Sigma$-norm_, and thus learns the space spanned by the $\Delta_t w_t$. We visualize this exact setting in Section B.1.1 in the appendix, and further elaborate on this intuition.
> > >
> > > We would be happy to address any other lingering questions/concerns the reviewer may have.

---

### Official Review · Reviewer_h1FC · 2021-08-02

**Rating:** 6
**Confidence:** 3

**Summary:**

The authors present a theoretical framework for analyzing the sample complexity of finetuning using an initialization from a MAML-like algorithm. The authors presented statistical analysis of fine-tuning-based meta-learning. They provide risk bounds on predictors found by fine-tuning via gradient descent. They demonstrate the success of such algorithms under the assumption of approximately shared representations between available tasks. They also establish settings where learning one representation for all tasks fails. They show that “frozen representation” objectives analyzed by prior work fail under this weaker assumption.

**Limitations And Societal Impact:**

Yes

**Main Review:**

The paper is well written and structured. The theoretical results could be useful for practitioners, such as fine-tuning quickly adapts to new tasks, requiring fewer samples in certain cases compared to using “frozen representation”. It is great to see the authors provide both linear and nonlinear analysis. In the linear case, they show there is a hard case that “frozen representation” fails to capture the shared structure from source tasks. In the nonlinear setting, they provide both linear regression and a two-layer network.

However, the paper does not have any empirical verification. I would hope the theoretical result can bring more insight to the questions in practice:
1) In the few short regimes, the frozen representation is usually preferred over end-to-end finetuning (AdaptiveRep) due to the concern of overfitting. With the theoretical result, how can we determine when to choose FrozenRep or AdaptiveRep?
2) Ideally the more similar between the source task and the target task, the better the fine-tuning performance. Can the performance bound predict the generalization ability for different tasks?
3) The linearity approximation in the “target loss landscape” (sec 4.2) seems to be not quite realistic. Previous practical work [1] have shown that end-to-end finetuning or adaptiveRep is often preferred when the target domain is not quite similar to the source domain and larger learning rate is preferred, which makes the linearity assumption invalid.

Some claims can be made clearer, e.g., in line 43 the author claims that “no prior studies exist beyond fine-tuning a linear model or only the task-specific layers”. The authors might want to clarify that it refers to theoretical studies but not practical works. Fine-tuning pre-trained networks in an end-to-end fashion has been common in practice for transfer learning. It would be great to also clarify the difference with previous approaches in the related work section.


[1] Li et al, Rethinking the Hyperparameters for Fine-tuning, ICLR 2020

**Time Spent Reviewing:**

6

---

> ### Author Response · Authors · 2021-08-10
> **Response to Reviewer h1FC**
>
> We thank the reviewer for their review, and appreciate their encouraging comments regarding our technical contributions and their potential to inform practice. We address the reviewer's concerns in order:
>
> > However, the paper does not have any empirical verification.
>
> We refer the reviewer to Section B of the appendix, where we included simulations comparing AdaptRep and FrozenRep within our constructed “hard case”. These experiments corroborate our theoretical results.
>
> > In the few short regimes, the frozen representation is usually preferred over end-to-end finetuning (AdaptiveRep) due to the concern of overfitting. With the theoretical result, how can we determine when to choose FrozenRep or AdaptiveRep?
>
> The result indeed reflects standard intuitions of when one method is preferred over the other. Recall the rate $r_T$ defined between lines 192-193. The subspace component is learned at rate $O(k/n_T)$ by both methods, while AdaptRep learns the task-specific residual at rate $O(\delta_0/\sqrt{n_T})$. If $n_T$ is very small and $\delta_0$ is large, then FrozenRep will have a better rate, consistent with what is observed in practice.
>
> However, this assumes that FrozenRep is not incurring a large irreducible loss by not fine-tuning. Our hard case provides a representative example of when this large loss exists (i.e. AdaptRep is better). In particular, following the discussion in Section B.1.1 in the appendix, FrozenRep learns an incorrect representation if the predictive signal from the base representation is dominated by that of the fine-tuning component. Intuitively, this is because FrozenRep seeks a subspace capturing the most the predictive signal averaged across tasks.
>
> > Ideally the more similar between the source task and the target task, the better the fine-tuning performance. Can the performance bound predict the generalization ability for different tasks?
>
> Recall that $\delta$ is a bound on the task-specific residual for each task, and thus $\delta$ quantifies task similarity, including distance from source to target. In particular, larger $\delta$ implies more dissimilar tasks. Indeed, in the provided rates, the generalization error bound shrinks with $\delta$.
>
> > The linearity approximation in the “target loss landscape” (sec 4.2) seems to be not quite realistic. Previous practical work [1] have shown that end-to-end finetuning or adaptiveRep is often preferred when the target domain is not quite similar to the source domain and larger learning rate is preferred, which makes the linearity assumption invalid.
>
> We want to reemphasize that our analysis in Section 4.2 does use end-to-end finetuning, i.e. we perform gradient updates on all the parameters.
>
> Furthermore, Mu et. al [1] empirically demonstrate using the gradients as features in meta-learning works well, which corresponds to our approximate linearity assumption. In their experiments, they demonstrate that using gradients to approximate fine-tuning is competitive/achieves better results on several standard benchmarks.
>
> > Some claims can be made clearer, e.g., in line 43 the author claims that “no prior studies exist beyond fine-tuning a linear model or only the task-specific layers”. The authors might want to clarify that it refers to theoretical studies but not practical works. Fine-tuning pre-trained networks in an end-to-end fashion has been common in practice for transfer learning. It would be great to also clarify the difference with previous approaches in the related work section.
>
> We will also work to more properly qualify our claims of novelty as being with respect to theoretical works, and provide more context comparing our overall approach with prior empirical work. To summarize, we provide a theoretical analysis for the commonly used end-to-end finetuning of pre-trained networks, and demonstrate the advantage of end-to-end training over only training the task-specific head.
>
> We hope that the above addresses the reviewer’s concerns, and that the reviewer will increase the score if our explanations are satisfactory. We will incorporate the discussion points above to improve the clarity of our work, and we would be happy to further elaborate/respond to any other concerns that the reviewer may have during the rolling discussion period!
>
> [1] Mu, F., Liang, Y., & Li, Y. (2019, September). Gradients as Features for Deep Representation Learning. In International Conference on Learning Representations.

---

### Decision · Program_Chairs · 2021-09-28

**Decision:**

Accept (Poster)

**Comment:**

This paper provides some theoretical insight into when and why fine-tuning is beneficial for meta-learning. Specifically, it shows how learning a single fixed representation for all tasks can fail, and under what conditions allowing the representation to be tuned for downstream tasks can be successful. Reviewers all appreciated the addition of theoretical insight and a framework for understanding meta-learning. The main criticism from reviewers was that it was hard to get intuition for the assumptions made and the settings considered. Authors should include better examples and diagrams in the camera-ready version.

**Consistency Experiment:**

NeurIPS has a long history of experimentation. In 2014, NeurIPS ran an experiment in which 10% of submissions were reviewed by two independent committees to quantify the randomness in the review process. This year, we repeated a variant of this experiment to see how the quality of the review process has changed over time.  This paper was part of the experiment and was therefore assigned to two committees (consisting of reviewers, an Area Chair, and a Senior Area Chair) that reached independent decisions.  If both committees made the same recommendation, this recommendation was followed. If a single committee recommended acceptance, the paper was accepted (with the exception of a few cases in which the other committee identified what we considered a fatal flaw, e.g., an error in a key result).

Both committees reached the same decision: **Accept (Poster)**

The other committee assigned to the paper recommended **Accept (Poster)**.  You can find the other set of reviews, along with any follow up discussion with the authors here:
https://openreview.net/forum?id=4HDwT8l12UK